# Association of Multi-Phasic MR-Based Radiomic and Dosimetric Features with Treatment Response in Unresectable Hepatocellular Carcinoma Patients following Novel Sequential TACE-SBRT-Immunotherapy

**DOI:** 10.3390/cancers15041105

**Published:** 2023-02-09

**Authors:** Lok-Man Ho, Sai-Kit Lam, Jiang Zhang, Chi-Leung Chiang, Albert Chi-Yan Chan, Jing Cai

**Affiliations:** 1Faculty of Health and Social Sciences, The Hong Kong Polytechnic University, Hong Kong; 2Radiotherapy and Oncology Centre, Gleneagles Hospital Hong Kong, Hong Kong; 3Research Institute for Smart Ageing, The Hong Kong Polytechnic University, Hong Kong; 4Department of Biomedical Engineering, Faculty of Engineering, The Hong Kong Polytechnic University, Hong Kong; 5Department of Health Technology and Informatics, The Hong Kong Polytechnic University, Hong Kong; 6Department of Clinical Oncology, School of Clinical Medicine, The University of Hong Kong, Hong Kong; 7Department of Surgery, School of Clinical Medicine, The University of Hong Kong, Hong Kong

**Keywords:** magnetic resonance imaging, radiomics, trans-arterial chemoembolization, stereotactic body radiotherapy, immunotherapy, hepatocellular carcinoma

## Abstract

**Simple Summary:**

Hepatocellular carcinoma (HCC) is one of the most prevalent and devastating malignancies worldwide. An ongoing phase-II clinical trial assesses the efficacy of a novel sequential trans-arterial chemoembolization (TACE) plus stereotactic body radiotherapy (SBRT) plus immunotherapy strategy as an induction therapy for unresectable HCC patients. This study aims to investigate the potential association between radiomic features extracted from pre-treatment multi-phasic MR images and treatment response following the novel intervention strategy. In this study, Four DeltaP-derived radiomics that characterize the temporal change in intratumoral randomness and uniformity were identified as the contributors to the treatment response for a 3-month timepoint. Additional arterial phase (AP)-derived radiomic features and tumor morphology were also shown to have strong associations with treatment response for a 6-month timepoint. The success of this study would demonstrate the feasibility of pre-treatment identification of responsive HCC patients, paving the way toward effective and personalized oncology for HCC management.

**Abstract:**

This study aims to investigate the association of pre-treatment multi-phasic MR-based radiomics and dosimetric features with treatment response to a novel sequential trans-arterial chemoembolization (TACE) plus stereotactic body radiotherapy (SBRT) plus immunotherapy regimen in unresectable Hepatocellular Carcinoma (HCC) sub-population. Twenty-six patients with unresectable HCC were retrospectively analyzed. Radiomic features were extracted from 42 lesions on arterial phase (AP) and portal-venous phase (PVP) MR images. Delta-phase (DeltaP) radiomic features were calculated as AP-to-PVP ratio. Dosimetric data of the tumor was extracted from dose-volume-histograms. A two-sided independent Mann–Whitney U test was used to assess the clinical association of each feature, and the classification performance of each significant independent feature was assessed using logistic regression. For the 3-month timepoint, four DeltaP-derived radiomics that characterize the temporal change in intratumoral randomness and uniformity were the only contributors to the treatment response association (*p*-value = 0.038–0.063, AUC = 0.690–0.766). For the 6-month timepoint, DeltaP-derived radiomic features (n = 4) maintained strong clinical associations with the treatment response (*p*-value = 0.047–0.070, AUC = 0.699–0.788), additional AP-derived radiomic features (n = 4) that reflect baseline tumoral arterial-enhanced signal pattern and tumor morphology (n = 1) that denotes initial tumor burden were shown to have strong associations with treatment response (*p*-value = 0.028–0.074, AUC = 0.719–0.773). This pilot study successfully demonstrated associations of pre-treatment multi-phasic MR-based radiomics with tumor response to the novel treatment regimen.

## 1. Introduction

Hepatocellular carcinoma (HCC) is one of the most prevalent and devastating malignancies worldwide, ranking as the 4th leading cause of cancer-related deaths. It accounts for 80–90% of the sufferers of primary liver cancer, and its highest incidences occur in eastern and southeastern Asia and northern Africa [1,2]. Surgical resection and liver transplantation have been the gold standard curative therapies. Unfortunately, most HCC patients present intermediate to advanced disease at diagnosis [3]. As such, more than 70% of liver cancer patients are considered ineligible for such curative interventions [4], partly due to the presentation of large-sized tumors, poor liver function, or organ shortage. The median survival remains at approximately 16 months for intermediate-stage HCC patients, and half a year for advanced-stage HCC patients, respectively [5], reflecting grievous survivorship in this vulnerable HCC sub-population.

Over the past decades, three key additional regimens have been developed in the hope of serving either as a bridging therapy before liver transplantation or as a curative alternative for unresectable HCC patients; they are Trans-Arterial Chemoembolization (TACE), Stereotactic Body Radiotherapy (SBRT) and Immune Checkpoint Blockade (ICB).

TACE has been widely adopted as a first-line treatment for intermediate-stage HCC [6]. It works by interrupting the major source of oxygen and nutrition supply to the cancer cells from the hepatic arteries, meanwhile selectively delivering cytotoxic chemotherapeutic agents for cancer eradication [7]. However, its efficacy is limited in patients with poor baseline liver function and larger tumor burden, hence TACE alone is often not sufficient for thorough cancer cell elimination in advanced-stage HCC patients [8]. On the other hand, SBRT kills cancer cells non-invasively by delivering an ultra-high radiation dose in a few fractions (usually ≤ 5) to the tumor in a highly precise and conformal manner, under real-time liver and tumor motion monitoring [9]. The survival benefits of SBRT in HCC have been well-documented in the literature for early stage tumors [10,11]. Recently, efforts have been made to investigate the efficacy of sequential TACE-SBRT in intermediate and advanced-stage HCC on the grounds of the reported potential synergism between TACE and radiotherapy [12,13,14,15,16,17,18]. For instance, Chiang et al. reported a promising efficacy of combined TACE-SBRT treatments in Barcelona Clinic Liver Cancer (BCLC) system stage B-C HCC patients, yielding an objective response rate of 68% and a 1-year local control rate of 93.6% [19]. However, intra-hepatic and distant dissemination remains a key challenge for managing this subgroup of unresectable HCC patients. Sequential SBRT-immunotherapy has demonstrated improved local tumor control and distant abscopal effect [20], potentially compensating for the deficiency of the sequential TACE-SBRT regimen. Our pilot studies have shown the satisfactory efficacy and safety of combined SBRT and immunotherapy for HCC patients [21,22].

Notably, for the first time in history, there is an ongoing phase-II clinical trial conducted by our group that aims to assess the efficacy of a novel sequential TACE-SBRT-Immunotherapy strategy as an induction therapy for unresectable HCC patients [23], the results are greatly anticipated. While exciting, the potential toxicities associated with this novel aggressive treatment are yet to be reported. In the era of personalized medicine, there is a pressing demand to discriminate between responders and non-responders prior to treatment commencement for the sake of avoiding ineffective and toxic therapies in non-responders and enhancing individualized oncologic care delivery.

The field of radiomics has been caught in the spotlight of attention within the medical community [24,25,26,27,28,29,30,31]. It involves high-throughput extraction of quantitative features from medical images for divulging intrinsic biological and genetic characteristics [32]. The capability of radiomics has been extensively reported in various cancer prediction tasks, including cancer prognosis [33], disease differentiation [34], and treatment response [35,36], highlighting the high potential of radiomics in informing decision-making in a wide spectrum of oncologic care. Apart from this, several research groups have reported improved predictive power when combining both radiomics and radiation dosimetric parameters [37,38]. Particularly, for HCC management, multi-phasic contrast-enhanced magnetic resonance (MR) images are routinely used for obtaining dynamic information on disease pathology and physiology. The role of multi-phasic MR-based radiomics in HCC has been widely studied for predicting micro-vascular invasion [39,40], cancer recurrence [41,42,43], disease diagnosis [44,45], and treatment response [46,47,48,49]. Nevertheless, radiomics studies on treatment response prediction in unresectable HCC patients are scarce [49]. Further, there is no study assessing the association between radiomic features extracted from pre-treatment multi-phasic MR images and treatment response following the novel sequential TACE-SBRT-Immunotherapy regimen.

In this pilot study, we aimed to investigate the association of pre-treatment multi-phasic MR-based radiomics and dosimetric features with treatment response to the sequential TACE-SBRT-Immunotherapy regimen in unresectable HCC sub-population, who were prospectively enrolled in the first-of-its-kind phase-II clinical trial [23]. The success of this study would demonstrate the feasibility of pre-treatment identification of responsive HCC patients for this novel regimen, paving the way towards effective and personalized oncology for HCC management worldwide in the long run.

## 2. Materials and Methods

### 2.1. Patient Data

The present study was approved by the Human Subjects Ethics Subcommittee of The Hong Kong Polytechnic University and Institutional Review Board of the University of Hong Kong/Hospital Authority Hong Kong West Cluster. Apart from this, patient data that were analyzed in this study were prospectively enrolled in an ongoing phase-II clinical trial conducted by The University of Hong Kong, entitled “Sequential TransArterial hemoembolization and stereotactic Radiotherapy Followed by ImmunoTherapy for downstaging hepatocellular carcinoma for hepatectomy (START-FIT)” [23]. A total number of 26 newly diagnosed HCC patients, who were treated with sequential TACE-SBRT-Immunotherapy at the Department of Clinical Oncology of Queen Mary Hospital (QMH) between May 2019 to October 2021, were retrospectively analyzed. The inclusion criteria included: (1) diagnosis of unresectable HCC confirmed pathologically according to the American Association for the Study of Liver Diseases (AASLD) practice guideline 2010; (2) male or female between 18 and 80 years old; (3) tumor size between 5 and 15 cm, and the number of lesions less than 3; (4) portal vein involvement; (5) Child–Pugh liver function class A-B7; (6) liver volume minus intrahepatic gross-tumor-volume (GTV) > 700 cc; (7) no prior TACE; and (8) no prior systemic therapy nor immunotherapy, TACE or RT. The specific contraindications of SBRT were: any HCC tumor >15 cm; total maximum sum of HCC diameter >20 cm; more than 3 discrete hepatic nodules; direct tumor extension into the stomach, duodenum, small bowel, large bowel, and main branch of biliary tree.

### 2.2. Treatment Details

SBRT was performed by using 6 MV or 10 MV photon beams delivered from a linear accelerator within 21–35 days after TACE. The prescribed dose ranged from 27.5 Gy to 50 Gy in 5 fractions, depending on normal tissue constraints. The time interval between fractions was limited to 24 to 72 h, with radiation delivered to all targets within 5 to 15 days. Varian External Beam Planning Software (Varian Medical Systems, Palo Alto, CA, USA) was used for treatment planning. The dosing scheme aimed at using the highest allowable prescription dose for the tumor target, while fulfilling the constraints of surrounding organs-at-risk (OARs). TACE and immunotherapy procedures were performed according to the routine treatment protocol in QMH. For TACE, an emulsion of a mixture of cisplatin with lipiodol in a volume ratio of 1 to 1 was prepared and injected into the tumor by femoral artery puncture. The amount of TACE used was based on tumor size, number, and arterial blood flow. For immunotherapy, Avelumab was administered via IV injection two weeks after SBRT; the amount of dose required was based on the patient’s body weight and toxicity.

### 2.3. Clinical Endpoint

The clinical endpoint of this study was defined as the response rate at 3 and 6 months after SBRT, according to the modified Response Evaluation Criteria in Solid Tumors ((mRECIST) version 1.1) criteria. The response rates were categorized into: (1) Complete response (CR) that represents the disappearance of any intratumoral arterial enhancement in all target lesions; (2) partial response (PR) that represents at least a 30% decrease in the sum of diameters of viable target lesions; (3) stable disease (SD) that represents any cases that do not qualify for either partial response or progressive disease; and (4) progressive disease (PD) that represents an increase of at least 20% in the sum of the diameters of viable (enhancing) target lesions. In this study, the treatment response of each lesion was assessed by a radiologist with 15 years of experience. Prior to subsequent analysis, patients with (1) CR and (2) PR were grouped into a respondent group, while those with (3) SD and (4) PD were grouped into a non-respondent group.

### 2.4. MRI Acquisition and Segmentation

The pre-intervention gadoxetic acid-enhanced MRI was obtained by using either 1.5T GE Signa system (version: HD16. GE Healthcare, Milwaukee, WI) or Philips 3T MRI Achieva scanner (Philips Healthcare, Best, The Netherlands) with a 12 or 16 channel, phased-array body coil. The image sets were acquired according to the START-FIT and LI-RADS ver. 2017 protocol, including axial arterial phase (AP) and portal venous phase (PVP) T1W image sets. A demonstrative example of AP and PVP MRI is shown in Figure 1a,b.

The segmentations (including gross tumor targets and OARs) were manually delineated on the axial planning CT slice-by-slice by an experienced clinical oncologist (with >15 years of experience). The contours of the lesions were subsequently transformed into other image sets by rigid registration for further processing. The transferred tumor lesions on the pre-intervention MR image sets were defined as the volumes of interest (VOIs).

### 2.5. Dosimetric Features

Dosimetric features of each lesion were obtained from the dose-volume histograms (DVHs) using the treatment planning system, including volume of GTV and planning-target-volume (PTV), prescription dose, minimum and maximum dose, mean dose, relative GTV volumes (in percentage) receiving specific doses (V5 to V50 in 5 Gy increments), minimum doses to relative liver volumes (D10% to D90% in 10% increments). In total, 33 dosimetric features were calculated for each lesion. A demonstrative example of DVH and dose distribution is shown in Figure 1c,d.

**Figure 1 cancers-15-01105-f001:**
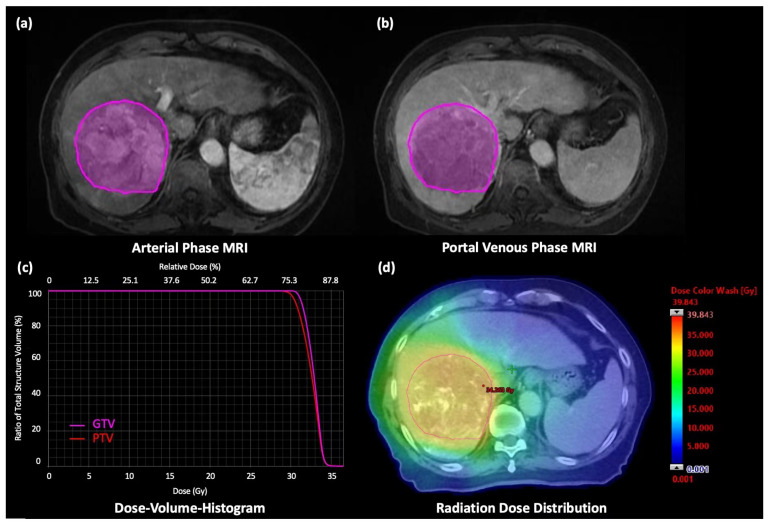
A 76-year-old male patient was diagnosed with advanced-staged HCC. (**a**) Axial AP T1W MR image with the VOI, (**b**) Axial PVP T1W MR image with the VOI. SBRT of 30 Gy was prescribed in 5 fractions to the tumor. (**c**) DVH of gross tumor volume (GTV) and planning target volume (PTV) generated from the treatment planning system and (**d**) Dose distribution of the SBRT treatment plan.

### 2.6. Image Preprocessing and Radiomic Features Extraction

The MR images with VOIs were imported into a python-based pipeline developed by The Hong Kong Polytechnic University, which was employed previously by other studies [50,51,52,53]. Before extracting the radiomics features, multiple pre-processing steps were performed. In order to tackle the parameter variations between image series, isotropic resampling was performed by linear interpolation to obtain a 1 × 1 × 1 mm^3^ voxel size. Inhomogeneity correction was performed by the N4 bias field correction algorithm to correct the locally varying intensity. Image intensities were then normalized by shifting and rescaling each image into a mean of zero and a standard deviation of 100 to maintain consistent voxel values across patients. They were further discretized by a fixed bin count of 100 to reduce the noise of image textures.

In this study, radiomics features were extracted from each lesion on both arterial phase (AP) T1W and portal venous phase (PVP) T1W MR images using the Pyradiomics (version 2.2.0) package. Delta phase (DeltaP) radiomics features were calculated by dividing feature values extracted from AP images by those extracted from PVP images. Multiple types of Radiomics features were extracted, including shape and size features (n = 14), first-order features (n = 18), and second-order texture features (n = 73). The texture features were calculated from gray-level co-occurrence matrices ((GLCMs): n = 22), gray-level run-length matrices ((GLRLMs): n = 16), gray-level size zone matrix (GLSZM: n = 16), gray-level dependence matrices ((GLDMs): n = 14), and neighboring gray tone difference matrices ((NGTDMs): n = 5). In total, 287 radiomic features were calculated for each lesion. The meaning of each radiomic feature parameter for Pyradiomics can be found in the link: https://pyradiomics.readthedocs.io/en/latest/features.html.

### 2.7. Statistical Analysis

For each of the studied endpoints, a two-sided independent Mann–Whitney U test was employed to assess the clinical association between treatment response (i.e., respondent or non-respondent) and features in each of the studied feature categories (i.e., AP, PVP, DeltaP radiomic features, and dosimetric features). Features with *p*-values of <0.05 were considered statistically significant. Moreover, independent endpoint-associated features were identified by using multiple feature selection procedures for each feature category. Ten independent features were first selected by using K-Means clustering from the original feature sets with a cluster number of ten. Cluster centroids were randomly initialized with 100 iterations to reduce potential bias. Independent features were then identified by the most significant features from the Mann–Whitney U test within each cluster. *p*-values were adjusted with false discovery rate (FDR) by using Benjamini–Hochberg (BH) multiple test correction among the identified ten independent features. Features with an FDR-adjusted *p*-value of <0.1 were reported. To further assess the predictability of each significant independent feature, logistic regression analysis was applied; the area under the receiver operator characteristic (ROC) curve (AUC), sensitivity, and specificity were then reported. Prior to regression analysis, all features were re-scaled to a mean of 0 and a standard deviation of 1.

Apart from this, a two-sided independent student *t*-test was used to determine whether there existed a statistically significant difference in patient demographic variables between the respondent and non-respondent groups. All statistical analyses were implemented using R software (version 4.2.1. The R Foundation, Vienna, Austria) and SPSS 26.0 (IBM, Chicago, IL, USA).

## 3. Results

### 3.1. Patient Characteristics

Table 1 summarizes patient characteristics. In total, 26 patients (male/female: 25/1; mean age: 67 ± 7.6 years) were included in the present study. Among the 26 patients, 14 had a single lesion, 8 had two lesions, and 4 had three lesions, resulting in a total of 42 lesions. The average diameter of the largest tumor nodule was 9.4 ± 3.7 cm.

### 3.2. Lesion Characteristics and Response Rate

Table 2 displays the overall characteristics and response rates of the lesions in respondent and non-respondent groups in both studied timepoints. For the 3-month assessment, respondents were identified in 18 lesions (42.9%, CR: n = 3; PR: n = 15), while non-respondents were identified in 24 lesions (57.1%, SD: n = 16; PD: n = 8). No statistically significant difference in GTV and PTV volumes was observed between groups. For the 6-month assessment, respondents and non-respondents were identified in 28 lesions (66.7%, CR: n = 13; PR: n = 15) and 14 lesions (33.3%, SD: n = 6; PD: n = 8), respectively. There were significant differences between respondent and non-respondent groups in average GTV volumes (295.4 ± 376.2 cm^3^ vs. 732.2 ± 728.2 cm^3^, *p* = 0.014) and average PTV volumes (418.4 ± 479.6 cm^3^ vs. 940.4 ± 857.0 cm^3^, *p* = 0.015).

### 3.3. Clinical Associations between Features and Treatment Response

Table 3 summarizes the statistically significant features identified from the two-sided independent Mann–Whitney U test for 3-month and 6-month response rate, respectively. Based on the results from Table 3, there was an inclination that different modalities of the pre-treatment multi-phasic MR images (AP, PVP, DeltaP) contain specific types of radiomic predictors associated with the response rate in HCC patients treated by the sequential TACE-SBRT-Immunotherapy regimen.

For 3-month response rate assessment, as shown in Table 3a, a total of 17 radiomic features were found to have a significant association with the 3-month response rate (*p* < 0.048), mostly uniformity- and entropy-related features (n = 13/17, 76%). Moreover, DeltaP-derived radiomic features accounted for the largest proportion (n = 15/17, 88%), followed by PVP-derived radiomic features (n = 2/17, 12%). However, it is worth noting that no shape and size features, dosimetric features, and AP-derived radiomic features were found to be significantly different between the respondent and non-respondent groups.

For the 6-month response rate assessment, as indicated in Table 3b, a total of 34 features (Radiomic features: n = 31; Shape and size features: n = 2; Dosimetric features: n = 1) were found to demonstrate significant clinical associations (*p* < 0.049). Among the 31 radiomic features, AP-derived radiomic features accounted for the largest proportion (n = 16/31, 52%), which was absent for the 3-month timepoint (Table 3a); this was followed by DeltaP-derived (n = 14/31, 45%), and PVP-derived (n = 1/31, 3%) radiomic features. Similar to the results from the 3-month response rate assessment, uniformity- and entropy-related features dominated (n = 13/31, 42%), especially in DeltaP-derived features (n = 9/14, 64%). Intriguingly, a considerable proportion of high gray level emphasis-related features (n = 8/31, 25%) demonstrated a statistically significant association with the 6-month response assessment, particularly in AP-derived features (n = 7/16, 43%).

Table 4 showcase a list of independent predictors that were determined to demonstrate the significant clinical association with the 3-month and 6-month response rate assessment, respectively. Based on the results from Table 4, previous findings of the inclination shown in Table 3 remained consistent and valid.

For the 3-month response rate assessment, 4 DeltaP-derived radiomic features were determined as independent predictors (FDR-adjusted *p*-value < 0.1, ranging from 0.038 to 0.063), as shown in Table 4a. The 4 predictors were mostly uniformity- and entropy-related features (GLCM_Joint Entropy, GLRLM_Run Entropy, GLSZM_Gray-Level Non-Uniformity Normalized and GLSZM_Small Area Emphasis), yielding AUC between 0.690 and 0.766. On the other hand, no shape and size features, dosimetric features, AP-derived, and PVP-derived radiomic features were identified as independent predictors (Table 4a).

For the 6-month response rate assessment, 9 features (Radiomic features: n = 8; Shape and size features: n = 1) were determined as independent predictors (FDR-adjusted *p*-value < 0.1, ranging from 0.028 to 0.074), as shown in Table 4b. Among the radiomic predictors, 4 were DeltaP-derived radiomic features, in which 3 were uniformity- and entropy-related features (GLCM_Joint Entropy, GLRLM_Run Entropy, GLSZM_Gray-Level Non-Uniformity Normalized), yielding AUC between 0.699 and 0.788. On the other hand, 4 radiomic predictors were derived from AP MR images, in which 2 were High Gray Level Emphasis-related radiomic features (GLRLM_Short Run High Gray-Level Emphasis, GLDM_Small Dependence High Gray-Level Emphasis), yielding AUC between 0.719 and 0.773. Moreover, a shape feature of Major Axis Length of the HCC lesions was selected as an independent predictor (FDR-adjusted *p*-value = 0.074), with an AUC of 0.724. No PVP-derived radiomic features and dosimetric features were identified as independent predictors (Table 4b).

## 4. Discussion

Unresectable HCC patients present a vulnerable sub-population of liver cancer patients. Over the years, different forms of combined sequential treatments, such as the TACE-SBRT and SBRT-Immunotherapy have been investigated as either a curative alternative or a bridging therapy for subsequent liver transplantation. For the first time in history, a novel sequential TACE-SBRT-Immunotherapy is being introduced in the hope of integrating the potential synergisms among these three treatment modalities, for this HCC subgroup in an ongoing phase-II clinical trial [23]. While exciting, the clinical benefits may be restricted only to a small portion of patients, especially when it comes to immunotherapy [54,55]. In the era of personalized medicine, there is a tremendous demand for pre-treatment discrimination between responsive and non-responsive candidates in order to avoid ineffective and toxic therapies in non-respondents.

In this pilot study, we successfully identified four independent predictors (All were DeltaP-derived radiomics) for 3-month response rate (FDR-adjusted *p*-value < 0.1, ranging from 0.038 to 0.063, Table 4a), and nine (four were DeltaP-derived; four were AP-derived; and one HCC shape feature) for 6-month response rate (FDR-adjusted *p*-value < 0.1, ranging from 0.028 to 0.074, Table 4b) to the sequential TACE-SBRT-Immunotherapy regimen in this prospectively enrolled unresectable HCC sub-population, paving the way towards effective and safe oncologic care delivery in the long run.

Results of the present study underscored that different modality of the pre-treatment multi-phasic MR-based radiomics (AP, PVP, and DeltaP) appears to contain specific types of textural predictors associated with the response rate in unresectable HCC patients treated by this novel aggressive regimen (Table 3 and Table 4). Specifically, the DeltaP-derived uniformity-related and entropy-related radiomic features were significantly associated with treatment response rate at both 3-month and 6-month timepoints; and the AP-derived high gray level emphasis-related radiomic features demonstrated a significant clinical association, particularly at the 6-month timepoint (Table 3 and Table 4).

The four AP-derived radiomic features that emerged to demonstrate significant association for 6-month response rate (“GLRLM_Short Run High Gray-Level Emphasis”, “GLDM_Small Dependence High Gray-Level Emphasis”, “Kurtosis”, “Maximum”) were related to the hyperintense signal intensity (i.e., the arterial-enhanced signal) on the AP image (Table 4b). “GLRLM_Short Run High Gray-Level Emphasis” measures the joint distribution of the short homogeneous runs with high gray-level, with a higher value indicating a greater concentration of high gray-level values within the lesion; “GLDM_Small Dependence High Gray-Level Emphasis” reflects the joint distribution of small dependence with higher gray-level values within the lesion; “Kurtosis” is a first-order statistics that informs the ‘peakedness’ of the distribution of values within the entire lesion, with a higher value implicating that the mass of the distribution is concentrated towards the tails; and another first-order measure of “Maximum” that tells the maximum gray-level intensity within the lesion. Although the biological meaning of these features in the context of HCC remains to be fully elucidated, they are all related to characteristics and spatial distribution of hyperintense signals within the lesion on AP images. This can be possibly ascribed by the fact that AP hyper-enhancement of tumor is a crucial property of HCC lesion [56], which reflects the tumor’s capability in generating new blood vessels (termed as neo-angiogenesis), an ability to drive more nutrition and oxygen supply exclusively from the hepatic arteries. In fact, “Kurtosis” has been frequently correlated to the response rate in various cancer types. For instance, Hou et al. conducted a radiomic study for the prediction of tumor response following systemic treatment of chemoradiotherapy in patients with esophageal carcinoma and reported that “Kurtosis” was identified as one of the most dominant features in their combined radiomic models for PR and CR prediction [57]. Moreover, Wang et al. applied radiomics for predicting tumor response to systemic induction chemotherapy in patients with nasopharyngeal carcinoma and reported “Kurtosis” as one of the predictive biomarkers in their MR-based prediction models [58]. Although this study presents a novel sequential TACE-SBRT-Immunotherapy, we speculated that these radiomic features characterizing baseline arterial-enhanced signal may be indicative of HCC responsiveness to treatment perturbations and deserve further in-depth investigations in the future.

Previous studies on multi-phasic MR-based radiomics for tumor response prediction in HCC patients have mainly focused on TACE treatment [47,48,49], while those on SBRT and immunotherapy regimens are scarce or absent. For TACE, Kuang et al. conducted a retrospective radiomics study to predict the response of small-sized HCC lesions following TACE and reported 11 AP-derived and 11 T2-weighted radiomic features as final predictors [47]. More recently, Liu et al. examined the predictive power of multi-phasic MRI for predicting HCC response following TACE treatment, and reported that 17 AP-derived and three PVP-derived radiomic features, along with radiomic features from other MR sequences [48]. However, these studies present a varying degree of disparity in study design compared with the present work in terms of target population [47], treatment regimen [47,48], and source images of radiomic features [47,48]. Therefore, a direction comparison of textual predictors between their studies and the present work appears to be infeasible and offers little scientific significance. Nonetheless, it is worth noting that the shape features of HCC seem to be predictive of treatment response despite the mentioned heterogeneity in study design between studies. In this study, the “Maximum 3D Diameter” was also found to demonstrate a significant association with treatment response to the sequential TACE-SBRT-Immunotherapy in the 6-month timepoint (Table 3b). Moreover, the “Major Axis Length”, another measure of baseline tumor burden which measures the largest axis length of the HCC-enclosing ellipsoid, was determined as an independent predictor (Table 4b). This finding is in line with the two previous studies where the “Maximum 3D Diameter” [47] and “Maximum 2D Diameter Row” [48] were found to be predictive of TACE treatment. Indeed, this result is also in concordance with the dynamic-CT-based study conducted by Park et al., where smaller tumor size was a significant predictor for complete response in HCC patients following TACE treatment [59].

On the other hand, Delta radiomics is a novel concept that reflects the dynamic variation in radiomic features in longitudinal images, highlighting intratumoral changes in imaging features and hence implicating the underlying tumoral physiological function. Compared with non-Delta radiomic features, Delta radiomics has recently been found to demonstrate higher reproducibility between scanners and institutions in phantom studies, potentially providing a more generalizable predictive capability, which has gained increasing popularity in the research community [60,61]. Nardone et al. provided a comprehensive systematic review of the Delta radiomics studies in the body of literature [60]. In the context of treatment response prediction, it has demonstrated ground-breaking evidence in numerous types of cancer following immunotherapy, including but not limited to renal cell cancer [62], pancreatic cancer [63], and non-small cell lung cancer [64,65,66,67]. Notably, to our best understanding, this study is one of the very first to report the potential of MR-based Delta radiomics in associating with treatment response in HCC.

Intriguingly, four DeltaP-derived radiomic features that were selected as independent significant predictors (Table 4b) were all related to randomness and uniformity of the spatial distribution of texture within the lesion. They were “GLCM_Joint Entropy” which measures randomness in signal intensity in neighboring voxels; “GLRLM_Run Entropy” which measures randomness in the distribution of run lengths and gray levels, with a higher value reflecting greater textual heterogeneity; “GLSZM_Gray-Level Non-Uniformity Normalized” which depicts the similarity of normalized gray-level intensity, with a lower value correlating with a greater similarity; and “Range” which tells the range of signal intensities within the lesion. These textual reflect the temporal change in the randomness and uniformity within the lesion between the AP and PVP MR scans, providing a better understanding of the intratumoral heterogeneity in time dimension upon arrival of the imaging contrast agent. Hence, these textual predictors may be indicative of the aggressiveness of HCC lesions and their responsiveness upon treatment perturbation. In fact, the radiomic features of “entropy” and “uniformity” have been well-recognized as of high prognostic value in various cancer types, including but not limited to HCC [68,69], esophageal cancer [70], lung cancer [71], and squamous cell carcinoma of head and neck cancer [72]. For instance, Liu et al. selected “Gray-Level Non-uniformity” as one of the radiomic predictors, which had the largest weight among other predictors, for overall survival prediction in HCC patients [68]. Another study conducted by Ganeshan et al. revealed that a higher hepatic entropy and lower uniformity often reflect a more complex tumor heterogeneity [73]. Along this line of thinking, Mulé et al. analyzed pre-treatment contrast-enhanced CT-based textures for overall survival prediction in advanced HCC patients following treatment with Sorafenib [69]. They reported a significant correlation between tumor heterogeneity and entropy at both AP and PVP phases, and particularly, the PVP-derived entropy was determined to be an independent prognostic factor [69]. These findings may partly explain why the four entropy- and uniformity-related features were determined to be of a significant clinical association with treatment response.

On the other hand, it is worth noting that the studied dosimetric data from the SBRT plan appeared to be not predictive of HCC treatment response prediction in this study. The dose parameter of V35Gy was identified as a significant predictor only for the 6-month timepoint under univariate analysis (Table 3b), which might imply that a dose threshold 35 Gy was required to trigger tumor responses. The biological effective dose for 35 Gy in five fractions was calculated as 59.5 Gy (α/β = 10), which was the minimum effective dose fractionation scheme mentioned in a systematic review and meta-analysis [74]. However, it was not shown predictive after multiple test corrections (Table 4b). In our study, SBRT was prescribed based on the isotoxic principle that the radiation dose was individualized based on tumor size, volume, and proximity of organ-at-risk. Such strategy was commonly adopted in treating large-sized, locally advanced HCC and often resulted in a heterogeneous dose [75,76]. Our study showed that the dose of 27.5–30 Gy was equally effective in terms of tumor response. The potential explanation is that immunotherapy may have sensitized the tumor to radiotherapy and that lower radiation doses can attain similar local control, as demonstrated in the pre-clinical study [77]. With this regard, it is interesting to note that while dosimetric parameters have been predictive mainly in the areas of toxicity prediction [37,38,78,79,80,81], there is scarce or none in treatment response prediction. The underlying reasons remain unknown, and it definitely represents an interesting research area for future scrutinization.

From a public health standpoint, the findings of this study demonstrated the feasibility of using cost-effective radiomics techniques in associating with treatment response in a highly vulnerable sub-population of HCC patients following a novel aggressive treatment. Patients in this subgroup often suffer not only from HCC but also other liver-related diseases, such as portal hypertension and ascites due to liver cirrhosis [82]. Taken together with the desperately poor survival rate, tremendous burdens have been placed on this patient subgroup and the healthcare system. Although the pioneering sequential TACE-SBRT-Immunotherapy regimen offers both local-regional and whole-body systemic therapy to this subgroup, the underlying toxicity profile remains unclear. In the long term, the results of this study may provide valuable insights into pre-treatment identification of responding and non-responding candidates for this novel treatment, so as to avoid ineffective, toxic, and costly therapies to refractory patients, while streamlining medical resourcing allocations within the healthcare system. Several limitations of this study should be acknowledged. First, the cohort of patients was small due to the stringent patient inclusion criteria for receiving the novel aggressive treatment in the prospective clinical trial. This limitation diminishes the strength of our results and prevents the possibility of using machine learning, AI algorithms or other sophisticated classification techniques for prediction model development [28]. Despite this, we were able to identify specific types of radiomic predictors from different multi-phasic MR images that can predict tumor response during sequential TACE-SBRT-Immunotherapy regimen, and provide classification performance at the individual feature level. More importantly, the key novelty of this present work lies in that we demonstrated the feasibility of using multi-phasic MR-based radiomics for predicting tumor response to the novel aggressive therapy in a vulnerable subgroup of HCC patients. Notably, patient recruitment in the clinical trial is continuously undertaken, and a larger-cohort study is anticipated and will be part of our future plan. Moreover, the reproducibility of radiomic features against tumor segmentation variability, and the correlation between radiomic features and genetic data were not investigated in this study. Moving forward with a larger cohort of patients, these should be considered when it comes to building robust predictive models for clinical use in the future.

## 5. Conclusions

In this pilot study, we successfully demonstrated that four DeltaP-derived radiomic features (characterizing temporal change in intratumoral randomness and uniformity), four AP-derived radiomic features (reflecting baseline tumoral arterial-enhanced signal pattern), and a tumor morphology (denoting initial tumor burden), were determined to be significantly associated with the 6-month response rate in unresectable HCC lesions following aggressive TACE-SBRT-Immunotherapy regimen, while the DeltaP-derived radiomics were the only contributors to the response rate at 3-month timepoint. While results indicated a potential for pre-treatment discrimination between responding and non-responding unresectable HCC candidates for this novel treatment, a larger study cohort is warranted in the future to validate the results of this work.

## Figures and Tables

**Table 1 cancers-15-01105-t001:** Patient characteristics.

Total Number of Patients	26
Gender	
Male	25
Female	1
Age (y), mean ± SD	67.0 ± 7.6
50–59	5
60–69	10
70–79	10
>79	1
Diameters of largest tumor nodule (cm), mean ± SD	9.4 ± 3.7
Sum of diameter of tumor nodule (cm), mean ± SD	12.6 ± 5.6
Medical History	
Hepatitis B	17
Hepatitis C	4
Alcoholics	2
Vascular invasion	
Portal vein involvement	6
Hepatic vein involvement	14
CP Score	
A5	19
A6	6
B7	1
BCLC Stage	
Stage A	3
Stage B	7
Stage C	16
Lesion numbers	
1 lesion	14
2 lesions	8
3 lesions	4

**Table 2 cancers-15-01105-t002:** Characteristics and response rates of the lesions in respondent and non-respondent groups in both studied timepoints.

	3 Months			6 Months		
	Respondent Group (n = 18)	Non-Respondent Group (n = 24)	*p*-Value	Respondent Group (n = 28)	Non-Respondent Group (n = 14)	*p*-Value
GTV size (cc) *, mean ± SD	374.3 ± 426.5	491.0 ± 634.6	0.504	295.4 ± 376.2	732.2 ± 728.2	0.014
<5 cc	3	3		4	2	
5–200 cc	5	11		12	3	
200–500 cc	5	1		6	1	
500–1000 cc	3	4		4	3	
>1000 cc	2	5		2	5	
PTV size (cc), mean ± SD	510.1 ± 534.9	654.1 ± 758.3.0	0.496	418.4 ± 479.6	940.4 ± 857.0	0.015
Prescribed Dose for SBRT, mean ± SD	34.3 ± 4.7	32.9 ± 5.2	0.377	33.9 ± 4.7	32.7 ± 5.5	0.449
27.5 Gy	1	4		2	3	
30 Gy	7	11		12	6	
35 Gy	4	3		5	2	
40 Gy	6	5		9	2	
45 Gy	-	1		-	1	
50 Gy	-	-		-	-	
Response Rate						
CR	3	-		13	-	
PR	15	-		15	-	
SD	-	16		-	6	
PD	-	8		-	8	

* GTV was defined as VOI in this study.

**Table 3 cancers-15-01105-t003:** (a) A list of statistically significant features identified from two-sided independent Mann–Whitney U test for 3-month treatment response assessment. (b) A list of statistically significant features identified from two-sided independent Mann–Whitney U test for 6-month treatment response assessment.

(a)	
Features	*p*-Value
** *PVP radiomic features* **	
*First-order feature*	
Uniformity	0.048
*Second-order feature*	
GLCM_Sum Entropy	0.040
** *DeltaP radiomic features* **	
*First-order feature*	
Entropy	0.045
Uniformity	0.029
*Second-order feature*	
GLCM_Difference Variance	0.011
GLCM_Joint Energy	0.040
GLCM_Joint Entropy	0.037
GLCM_Sum Entropy	0.033
GLRLM_Gray-Level Non-Uniformity	0.031
GLRLM_Gray-Level Non-Uniformity Normalized	0.037
GLRLM_Run Entropy	0.010
GLSZM_Gray-Level Non-Uniformity Normalized	0.010
GLSZM_Size Zone Non-Uniformity Normalized	0.003
GLSZM_Small Area Emphasis	0.003
GLDM_Dependence Entropy	0.029
GLDM_Gray-Level Non-Uniformity	0.019
NGTDM_Contrast	0.029
(b)	
**Features**	***p*-Value**
** *Shape and size features* **	
Major Axis Length	0.018
Maximum 3D Diameter	0.049
** *AP radiomic features* **	
*First-order feature*	
Kurtosis	0.008
Maximum	0.017
Uniformity	0.040
*Second-order feature*	
GLCM_Auto-correlation	0.004
GLCM_Joint Average	0.004
GLRLM_Gray-Level Non-Uniformity	0.049
GLRLM_Gray-Level Non-Uniformity Normalized	0.046
GLRLM_High Gray-Level Run Emphasis	0.004
GLRLM_Long Run High Gray-Level Emphasis	0.021
GLRLM_Short Run High Gray-Level Emphasis	0.004
GLSZM_Gray-Level Non-Uniformity	0.049
GLSZM_High Gray-Level Zone Emphasis	0.005
GLSZM_Small Area High Gray-Level Emphasis	0.005
GLDM_High Gray-Level Emphasis	0.004
GLDM_Small Dependence High Gray-Level Emphasis	0.021
NGTDM_Coarseness	0.049
** *PVP radiomic features* **	
*Second-order feature*	
GLSZM_High Gray-Level Zone Emphasis	0.046
** *DeltaP radiomic features* **	
*First-order feature*	
Entropy	0.012
Maximum	0.024
Median	0.049
Minimum	0.014
Range	0.038
Uniformity	0.030
*Second-order feature*	
GLCM_Joint Entropy	0.030
GLCM_Sum Entropy	0.013
GLRLM_Gray-Level Non-Uniformity	0.026
GLRLM_Gray-Level Non-Uniformity Normalized	0.030
GLRLM_Run Entropy	0.009
GLSZM_Gray-Level Non-Uniformity Normalized	0.002
GLDM_Gray-Level Non-Uniformity	0.002
NGTDM_Contrast	0.010
** *Dosimetric features* **	
V35Gy Percentage	0.035

**Table 4 cancers-15-01105-t004:** A list of independent predictors that demonstrated significant clinical association with the (a) 3-month and (b) 6-month response rate assessment. Statistical significance is indicated by false discovery rate (FDR)-adjusted *p*-value, after applying the Benjamini–Hochberg (BH) procedure for multiple test corrections. AUC, sensitivity, specificity, and *p*-value obtained from logistic regression for each of these independent significant predictors are also reported. The superscript ^a^ denotes FDR-adjusted *p*-values obtained after applying the BH procedure.

(a)				
Features	FDR-Adjusted *p*-Value ^a^	AUC	Sensitivity	Specificity
** *DeltaP radiomic features* **				
*Second-order feature*				
GLCM_Joint Entropy	0.063	0.690 (0.527–0.843)	0.625	0.667
GLRLM_Run Entropy	0.044	0.734 (0.573–0.869)	0.625	0.667
GLSZM_Gray-Level Non-Uniformity Normalized	0.038	0.734 (0.566–0.881)	0.625	0.667
GLSZM_Small Area Emphasis	0.038	0.766 (0.600–0.912)	0.625	0.667
(b)				
**Features**	**FDR-Adjusted *p*-Value ^a^**	**AUC**	**Sensitivity**	**Specificity**
** *Shape and size features* **				
Major Axis Length	0.074	0.724 (0.529–0.891)	0.714	0.607
** *AP radiomic features* **				
*First-order feature*				
Kurtosis	0.028	0.750 (0.589–0.895)	0.786	0.643
Maximum	0.028	0.727 (0.564–0.879)	0.714	0.607
*Second-order feature*				
GLRLM_Short Run High Gray-Level Emphasis	0.028	0.773 (0.606–0.917)	0.857	0.679
GLDM_Small Dependence High Gray-Level Emphasis	0.055	0.719 (0.533–0.859)	0.786	0.643
** *DeltaP radiomic features* **				
*First-order feature*				
Range	0.047	0.699 (0.518–0.865)	0.786	0.643
*Second-order feature*				
GLCM_Joint Entropy	0.070	0.707 (0.527–0.877)	0.643	0.571
GLRLM_Run Entropy	0.047	0.747 (0.593–0.883)	0.714	0.607
GLSZM_Gray-Level Non-Uniformity Normalized	0.047	0.788 (0.633–0.909)	0.786	0.643

## Data Availability

Not applicable.

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
