# Peer review of "Association of Multi-Phasic MR-Based Radiomic and Dosimetric Features with Treatment Response in Unresectable Hepatocellular Carcinoma Patients following Novel Sequential TACE-SBRT-Immunotherapy"

_cancers, 2023, doi:10.3390/cancers15041105_

Round 1

Reviewer 1 Report

This manuscript shows the effectiveness of combined TACE and SBRT treatment for HCC and its characteristics using MRI. Although the number of patients is not large, the content is well reviewed and organized. Some improvements are desired.

1. Please describe any inclusion and exclusion criteria especially for SBRT treatment. What are the cases in which it is not indicated due to tumor size, location, etc.?

2. It would be better to describe the significance and meaning of each imaging feature together in Table 4.

Reviewer 2 Report

I believe for radiation oncologist this paper is well written and will be highly received for

Radiation oncologist this is medical and surgical oncologist it slightly technical And it would be a challenge To completely understand

Reviewer 3 Report

I’m grateful for the opportunity of reviewing the article “Association of Multi-phasic MR-based Radiomic and Dosimetric Features with Treatment Response in Unresectable Hepatocellular Carcinoma Patients Following Novel Sequential TACE-SBRT-Immunotherapy” by Ho et al. The article focuses on the potential association between radiomic features extracted from pre-treatment multi-phasic MR images and treatment response following TACE-SBRT-Immunotherapy.

The article is well written and the language and style would require only minor spell checks.  

I would just point only 2 remarks:

  • SBRT prescrive doses are in a wide range (from 27,5 to 50 Gy in 5 fractions). As stated by the authors “it is worth noting that the studied dosimetric data from plan appeared to be not predictive to HCC treatment response prediction in this study”, but then they state that “35Gy in 5 fractions was calculated as 59.5Gy (α/β = 10), which was the minimum effective dose fractionation scheme mentioned in a systematic review”. How can they justify the adoption of 27,5 Gy in 5 fractions as effective dose in the study? 
  • In the “Image Peprocessing and Radiomic Features Extraction” the authors indicate that “The MR images with VOIs were imported into a python-based pipeline developed by The Hong Kong Polytechnic University.” Can they provide any other relevant information, such as previous publications which have validated this methodology, if available? 

Reviewer 4 Report

General comments: Well written manuscript regarding an interesting, relevant topic. 

Specific comments: 

Due to the large number of abbreviations used in this manuscript, a list of abbreviations after keywords may be helpful to the reader.

Page 31, line 65: Please reword sentence to "Median survival remains at approximately 16 months and half a year for immediate-stage and..."

Page 4, line 143: What does the acronym GTV denote? Please spell out each acronym the first time it is used.

Page 15 lines 443/444 state "this study is the first to report potential of MR-based delta-radionics with treatment response in HCC." I performed a search with Google scholar and PubMed which demonstrated the below article regarding this topic. Granted, this study only dealt with SBRT and not TACE or immunotherapy.  Therefore, this is one of the first articles, not the first.  It is a very small point that needs to be addressed.

Jin WH, Simpson GN, Dogan N, Spieler B, Portelance L, Yang F, Ford JC. MRI-based delta-radiomic features for prediction of local control in liver lesions treated with stereotactic body radiation therapy. Sci Rep. 2022 Nov 3;12(1):18631. doi: 10.1038/s41598-022-22826-5. PMID: 36329116; PMCID: PMC9633752.

Round 2

Reviewer 3 Report

Dear Editor,

thank you for the occasion to review an updated version of the manuscript "Association of Multi-phasic MR-based Radiomic and Dosimetric Features with Treatment Response in Unresectable Hepatocellular Carcinoma Patients Following Novel Sequential TACE-SBRT-Immunotherapy"
The authors did a good job in addressing my previously raised points.

Regards,